# Impact and cost-effectiveness of rotavirus vaccination in Niger: a modelling study evaluating alternative rotavirus vaccines

Frédéric Debellut ![ORCID],[1] Kevin Tang ![ORCID],[2,3] Andrew Clark,[4] Clint Pecenka,[5] Bachir Assao,[2] Ousmane Guindo,[2] Rebecca F Grais,[2] Sheila Isanaka[2,6]

¹Center for Vaccine Innovation and Access, PATH, Geneva, Switzerland
²Epicentre, Paris, France
³London School of Hygiene & Tropical Medicine, London, UK
⁴Health Services Research and Policy, London School of Hygiene and Tropical Medicine, London, UK
⁵Center for Vaccine Innovation and Access, PATH, Seattle, Washington, USA
⁶Departments of Nutrition and Global Health and Population, Harvard T.H. Chan School of Public Health, Boston, Massachusetts, USA

**Correspondence to**
Mr Frédéric Debellut;
fdebellut@path.org

## ABSTRACT

**Objectives** To evaluate the cost-effectiveness of alternative rotavirus vaccines in Niger, using UNIVAC, a proportionate outcomes model.

**Setting** The study leverages global, regional and local data to inform cost-effectiveness modelling. Local data were collected as part of a clinical trial taking place in the Madarounfa district, Maradi region, Niger.

**Participants** The study models impact of infants vaccination on rotavirus gastroenteritis in children under 5 years of age.

**Interventions** We compared the use of ROTARIX (GlaxoSmithKline, Belgium), ROTAVAC (Bharat Biotech, India) and ROTASIIL (Serum Institute, India) to no vaccination and to each other over a 10-year period starting in 2021.

**Results** We estimated that ROTARIX, ROTAVAC and ROTASIIL would each prevent 13 million cases and 20 000 deaths of children under 5 years over a 10-year period in Niger. Compared with no vaccination, the cost to avert a disability-adjusted life-year was US$146 with ROTARIX, US$107 with ROTASIIL and US$76 with ROTAVAC from the government perspective. ROTAVAC dominated ROTARIX and ROTASIIL (eg, provided similar or higher benefits at a lower cost) and had 90% chance to be cost-effective at a US$100 willingness-to-pay threshold.

**Conclusions** This study can inform decision-making around rotavirus vaccination policy in Niger, demonstrating that ROTAVAC is likely the most cost-effective option. Alternative products (ROTASIIL and ROTARIX) may also be considered by decision-makers if they are priced more competitively, or if their cold chain requirements could bring additional economic benefits.

## STRENGTHS AND LIMITATIONS OF THIS STUDY

⇒ This analysis uses a cost-effectiveness model (UNIVAC) developed to assist decision-makers in low-income and middle-income countries in evaluating vaccine policy options.
⇒ This economic evaluation leverages unique local data on cost to deliver rotavirus vaccines and cost borne by households for their children sick with rotavirus gastroenteritis.
⇒ Local data were collected in a single region of the country, and sensitivity analysis was performed to handle uncertainty around their representativeness for Niger.

## INTRODUCTION

Rotavirus remains the leading cause of diarrhoea mortality globally and was estimated to cause more than 150 000 deaths in children younger than 5 years of age in 2019, with more than 80% of these deaths occurring in sub-Saharan Africa.[1 2] The WHO recommends universal rotavirus vaccination for all children.[3]

Two live oral rotavirus vaccines (ROTARIX (GlaxoSmithKline, Belgium) and RotaTeq (Merck & Co, USA)) were prequalified and approved for global use by WHO in 2009.[4] One decade later, in 2018, two additional live oral rotavirus vaccines (ROTAVAC (Bharat Biotech, India) and ROTASIIL (Serum Institute, India)) entered the global market and received the same WHO prequalification status. These rotavirus vaccines are safe and effective and are used in more than 110 countries worldwide.[5 6]

In many of the world's poorest countries, including several African countries, rotavirus vaccines have been introduced with financial support from Gavi, the Vaccine Alliance. Gavi is an international organisation aiming to increase equitable and sustainable access to vaccines for low-income and middle-income countries. As a result, countries eligible for Gavi support can access negotiated prices from manufacturers through procurement with UNICEF. In 2014, the Republic of the Niger (hereafter referred to as Niger) introduced ROTARIX into its national immunisation programme with support from Gavi.[7] Prior to this, rotavirus was associated with around one-third of all severe diarrhoea cases

with moderate-to-severe dehydration in children aged under 5 years in the country.[8] The ROtavirus Safety and Efficacy (ROSE) clinical trial, implemented by Epicentre along with the Niger Ministry of Health and Médecins Sans Frontières, demonstrated the safety and efficacy of ROTASIIL in areas of Niger where ROTARIX was not yet available.[9 10] The ROSE clinical trial also collected data to estimate the cost of delivering rotavirus vaccine as part of the national immunisation programme as well as the costs borne by households when a child develops rotavirus illness.[10]

Despite inclusion of ROTARIX in the Expanded Programme on Immunisation (EPI) programme for several years and the availability of alternative rotavirus vaccines, there has been no formal economic evaluation of any rotavirus vaccine in Niger. This study aims to evaluate and compare the cost-effectiveness of ROTARIX, ROTAVAC and ROTASIIL using locally collected data. The results of this study can inform national decision-making on rotavirus vaccination policy in Niger.

## METHODS
### Study design
We estimated the health impact and cost-effectiveness of three different rotavirus vaccines in Niger from 2021 to 2030 from the government and societal perspectives. The government perspective includes healthcare costs supported by the government only, while the societal perspective also accounts for healthcare costs supported by households. We used an Excel proportionate outcomes model, UNIVAC (V.1.4.30), to compare three rotavirus vaccine products to no vaccination and to each other.[11] We evaluated the three rotavirus vaccines available through Gavi: ROTARIX, a monovalent human liquid vaccine available in a single-dose vial presentation and administered in a two-dose course; ROTAVAC, a monovalent human-bovine reassortant frozen liquid vaccine, available in a five-dose vial presentation and administered in a three-dose course; and ROTASIIL, a pentavalent human-bovine reassortant lyophilised vaccine available in a two-dose vial presentation and administered in a three-dose course.[3] We did not evaluate ROTATEQ as it is not available to Gavi-eligible countries, and therefore, would be unlikely considered for use in Niger by governmental authorities.[12]

The UNIVAC model was used to predict the number of averted rotavirus disease events in children under 5 years of age, related healthcare costs and vaccination programme costs. Input parameters were informed by data from the literature and primary data collected during the ROSE trial and its ancillary costing studies. The primary outcome measure is the incremental cost-effectiveness ratio (ICER) expressed as the cost in US$ per disability-adjusted life-year (DALY) averted. In the absence of a set willingness-to-pay (WTP) threshold in Niger, we interpret our cost-effectiveness results using 0.5 times the gross domestic product (GDP) per capita

(Niger's GDP per capita was US$553.9 in 2019[13]). All cost data are reported in 2021 USD, and we applied a standard 3% discount rate to all future costs and health outcomes.

### Parent trial and ancillary costing studies
The ROSE trial was a double-blind, placebo-controlled randomised phase III trial that was implemented in the Madarounfa district in Niger from August 2014 to February 2018 (ClinicalTrials.gov identifier: NCT02145000). It assessed the efficacy and safety of ROTASIIL vaccine administered at 6, 10 and 14 weeks of age against rotavirus disease in healthy infants, demonstrating the safety of the vaccine and an efficacy level of 54.7% (95% CI 38.1% to 66.8%) at 24 months of follow-up.[10 14] Two ancillary costing studies were carried out alongside the trial to collect data on (1) the costs of immunisation delivery in the national immunisation programme and (2) the costs of illness borne by households when they seek care for their child sick with rotavirus diarrhoea.

For costs of immunisation delivery, we included cost categories according to standard guidelines developed for the EPIC studies.[15] Primary data were collected from the central EPI at the national, regional and district levels, as well as at four health facilities. Data collected covered international handling and transportation costs, national, regional and district costs related to storing and distributing vaccines, the number of doses of vaccines stored and distributed, and the vaccines' presentation. These data were used to define a volume-based unit cost, which was then applied to each vaccine and their respective volume. We added costs at the health facility level, which was composed of labour costs to administer vaccines, storage and transportation costs. Details on costs included for immunisation delivery are available from online supplemental table S1. The main driver of cost was service delivery at health facility level, which represented between 30% and 60% of the total immunisation delivery cost, depending on which vaccine was considered.

Healthcare costs of rotavirus gastroenteritis (RVGE) illness are estimated from primary data collected within the ROSE trial as well as published estimates. For the government perspective, we used estimates of direct medical costs of diarrhoea as published by Baral et al.[16] For the societal perspective, direct medical costs were combined with household costs collected from a subgroup of caregivers presenting to ROSE trial health facilities with a child sick with RVGE.[17] Household costs included out-of-pocket expenditures to seek care in terms of medicines, transportation and food as well as indirect costs from time spent seeking care and caring for sick children. Caregiver time was valued using the annual GDP per capita divided by 365 days and multiplied by the number of days spent seeking care or caring for a child in hospital. Data were collected from February 2018 to April 2019 from caregivers presenting to an outpatient health facility (n=365) or whose child has been hospitalised (n=26). Data were collected using standard questionnaires used

**Table 1** Input parameters for estimation of disease burden

| Input parameters | Base case | Lower bound | Upper bound | References |
|---|---|---|---|---|
| Incidence (per 100 000 under-5 children per year) | | | | |
| Non-severe RVGE cases | 3260 | 2305 | 4701 | 10 18 |
| Non-severe RVGE visits | 1662 | 1176 | 2398 | 19 |
| Severe RVGE cases | 1992 | 1354 | 2550 | 10 18 |
| Severe RVGE visits | 1016 | 690 | 1301 | 19 |
| RVGE hospitalisations | 502 | 341 | 643 | 10 |
| RVGE Deaths | 115 | 74 | 176 | 22 |
| Duration of illness (days) | | | | |
| Non-severe RVGE case | 2.36 | 2.13 | 2.59 | 10 |
| Severe RVGE cases | 3.58 | 3.07 | 4.09 | 10 |
| Disability weights | | | | |
| Non-severe RVGE case | 0.188 | 0.125 | 0.264 | 23 |
| Severe RVGE cases | 0.247 | 0.164 | 0.348 | |
| Age distribution of disease events | | | | |
| Age distribution | Cumulative percentage | | | |
| < 1 month | 0% | | | 18 |
| < 2 months | 3% | | | |
| < 3 months | 10% | | | |
| < 6 months | 36% | | | |
| < 1 year | 80% | | | |
| < 2 years | 97% | | | |
| < 3 years | 99% | | | |
| < 4 years | 100% | | | |
| < 5 years | 100% | | | |

RVGE, rotavirus gastroenteritis.

to gather information on cost incurred by families during an episode.

All cost categories and line items included during this ancillary costing activities are available from online supplemental table S2.

### Rotavirus disease burden

We assumed that rotavirus infection would manifest as either non-severe RVGE or severe RVGE, aligning with classifications used in the trial. Non-severe RVGE resolves after seeking care in an outpatient setting or without seeking any care. Severe RVGE may resolve or be fatal, after seeking care in an outpatient or an inpatient setting or without seeking any care. Input parameters used for modelling rotavirus disease burden in Niger are available in table 1. To define incidence rates for the different disease events, we used the incidence of severe and non-severe RVGE from the ROSE trial placebo population (6 weeks to 2 years of age) and converted this to incidence among children aged <5 years using data on the age distribution of rotavirus hospital admissions in high-mortality settings.[18] We defined incidence of RVGE clinic visits using the proportion of mothers reporting seeking care

for their child sick with diarrhoea (51%) based the most recently published DHS survey in Niger.[19] We used ROSE trial data to inform the incidence of severe RVGE hospitalisations, using the proportion of children with severe RVGE that were brought to the hospital (25%). We also used the ROSE trial data to inform the duration of severe and non-severe RVGE episodes. Lower and upper bounds for incidence reflect the low and high estimates used for Niger in previously published multicountry analyses.[20 21] The rate of RVGE deaths was taken from Clark *et al*)[22] and represents the mean of three possible international estimates of RVGE mortality in Niger. DALY weights were taken from the Global Burden of Disease study, using moderate and severe diarrhoea as proxies for non-severe and severe RVGE.[23]

### Vaccine efficacy, coverage and coverage timeliness

Efficacy data for the three evaluated vaccines is taken from a meta-analysis pooling data from all published randomised controlled trials.[24] We assume that all products confer the same level of protection after the last dose of a full course, corresponding to 79% efficacy 2 weeks

after the last dose, with subsequent waning bringing efficacy to 45% 12 months after the last dose.

We assume that vaccine efficacy against non-severe RVGE was 85% of the efficacy assumed for severe RVGE.[25] In an alternative scenario, we assumed 0% vaccine efficacy against non-severe disease, as can be derived from the ROSE trial, when comparing the reported efficacy against severe RVGE and efficacy against RVGE of any severity.[10]

Vaccine coverage and timeliness data for rotavirus vaccine doses 1, 2 and 3 were assumed to be the same as vaccine coverage and timeliness reported for diphtheria, tetanus and pertussis (DTP) vaccine, where three doses of the vaccine are scheduled to be administered at 6, 10 and 14 weeks of age. For two-dose ROTARIX, we only applied coverage for DTP1 and DTP2. Vaccine timeliness data were based on analyses by Clark *et al*.[22 26 27]

### Vaccine prices and other procurement costs
We used vaccine prices published by Gavi, the Vaccine Alliance and assumed such prices would be available for the entire period of analysis.[28] In addition to benefitting from Gavi-negotiated prices, eligible countries only pay a share of vaccine procurement, dependent on their income per capita. In a scenario analysis, we applied the Niger cofinancing share as an initial self-financing country per Gavi policy.[29 30] Other vaccine procurement related costs include safety boxes and vaccine wastage.

Vaccine wastage accounts for vaccine doses that are discarded, lost, damaged or destroyed. Wastage is dependent on each vaccine presentation. We used wastage rates as recommended by Gavi's detailed product profiles and calculated with the WHO wastage rate tool. This resulted in a 4% wastage rate for ROTARIX, 5% for ROTAVAC, and 9% for ROTASIIL.[28 31] All Input parameters used for estimation of vaccine programme costs and healthcare costs are available from table 2.

### Sensitivity analyses
To account for uncertainty, we undertook deterministic and probabilistic uncertainty analyses. Deterministic analyses covered a series of scenarios more and less favourable to the vaccine. In scenario 1, we accounted only for the country cofinancing Niger currently pays for vaccine

**Table 2** Input parameters for estimation of vaccine programme costs and healthcare costs

| Input parameter | Base case | Lower bound | Upper bound | Sources |
|---|---|---|---|---|
| Vaccine coverage* | | | | |
| Dose 1 | 92% | 82.8% | 100% | 26 |
| Dose 2 | 86.5% | 77.9% | 95.2% | |
| Dose 3 (ROTAVAC and ROTASIIL only) | 81% | 72.9% | 89.1% | |
| Vaccine price per dose (US$) | | | | |
| ROTARIX | $2.33 | $0.20 | – | 28 |
| ROTAVAC | $0.85 | $0.13 | – | |
| ROTASIIL | $0.95 | $0.13 | – | |
| % Wastage | | | | |
| ROTARIX | 4% | 2% | 6% | 28 |
| ROTAVAC | 10% | 5% | 15% | |
| ROTASIIL | 9% | 5% | 15% | |
| Immunisation delivery cost per dose (2021 US$) | | | | |
| ROTARIX | $1.65 | $1 | $2.77 | ROSE costing study |
| ROTAVAC | $0.75 | $0.75 | $1.87 | |
| ROTASIIL | $1.20 | $1 | $2.31 | |
| Health care cost (2021US$) | | | | |
| Unit cost of RVGE visit | | | | 16 and ROSE costing study for households' costs |
| Government perspective | $4.79 | $2.40 | $7.19 | |
| Societal perspective | $7.16 | $4.77 | $9.56 | |
| Unit cost of severe RVGE hospitalisation | | | | |
| Government perspective | $18.38 | $9.19 | $27.57 | |
| Societal perspective | $28.68 | $19.49 | $37.87 | |

*For vaccine timeliness, we assume that by 6 months of age, 91% of children would have received their first dose, 80% would have received their second dose, and, when applicable, 60% would have received their third dose of rotavirus vaccine. Data were fitted using a gamma curve to estimate timeliness by week of age <5 years.
RVGE, rotavirus gastroenteritis.

procurement; in scenario 2, we accounted for a higher burden of disease; in scenario 3, we accounted for a lower burden of disease; in scenario 4, we accounted for a higher vaccine efficacy; in scenario 5, we assumed a lower vaccine efficacy; in scenario 6, we accounted for higher healthcare costs; in scenario 7, for lower healthcare costs; in scenario 8, for higher immunisation delivery costs and in scenario 9, we assumed vaccines had no efficacy against non-severe RVGE. Finally, as our assumptions around similar efficacy of the last dose of vaccine led to lower impact of the two-dose ROTARIX vaccine compared with three-dose ROTAVAC and ROTASIIL vaccines, we also evaluated the cost-effectiveness of ROTARIX with similar impact to the three-dose vaccines (scenario 10).

We ran a probabilistic uncertainty analysis for the three vaccines. We conducted 1000 model runs sampling input data and statistical distributions available in online supplemental table S3). We reported, for each vaccine, the mean ICER value from the government perspective as well as results for each model run. We also built a cost-effectiveness acceptability curve showing the probability for the dominant vaccine to be cost-effective at different WTP thresholds.

## No patient and public involvement

This research was done without patient or public involvement. It was not appropriate or possible to involve patients or the public in the design, or conduct, or reporting, or dissemination of our research as the study focused on assessing cost and cost-effectiveness of an intervention provided by the Ministry of Health.

## RESULTS
### Deterministic analysis

We estimate that a vaccination programme using any of the three rotavirus vaccines analysed could avert around 13 million RVGE cases and 20 000 deaths over 10 years in Niger. With 660 000 visits and 136 000 hospitalisations averted, rotavirus vaccination could result in healthcare cost savings of US$5 million from the government and US$7 million from the societal perspective. Vaccination programme costs vary depending on which vaccine is used: US$46.7 million with ROTAVAC, US$61.8 million with ROTASIIL and US$77.3 with ROTARIX, when not accounting for Gavi subsidies for vaccine procurement. Detailed figures on outcomes related to each vaccine are available in table 3. On average, undiscounted annual vaccination programme costs in absence of Gavi support vary between US$5.5 million with ROTAVAC and US$8.9 million with ROTARIX, with the majority of the cost dedicated to vaccines and supplies procurement (55% for ROTAVAC, 60% with ROTARIX).

The cost per DALY averted varies by vaccine and perspective. From the government perspective, and when comparing each vaccine to 'no vaccine', the most favourable ICER is with ROTAVAC (US$76 per DALY averted), followed by ROTASIIL (US$104) and then ROTARIX (US$146). From the societal perspective, the same order remains, with ROTAVAC having the lowest ICER (US$72), followed by ROTASIIL (US$99) and ROTARIX (US$141). When assuming two-dose ROTARIX had similar impact to the three-dose ROTAVAC or ROTASIIL, the ROTARIX ICER decreased to US$133 from the government perspective and US$128 from the societal perspective (online supplemental table S4).

ROTAVAC can be considered cost-effective based on a WTP threshold set at half the GDP per capita (US$277).

| Table 3 | Health and economic benefits of vaccine (2021–2030, discounted) | | |
| --- | --- | --- | --- |
| **Health and economic benefits** | **ROTARIX (two doses)** | **ROTAVAC (three doses)** | **ROTASIIL (three doses)** |
| Non-severe RVGE cases averted | 706 843 | 774 058 | 774 058 |
| Non-severe RVGE visits averted | 360 490 | 394 770 | 394 770 |
| Severe RVGE cases averted | 508 066 | 556 379 | 556 379 |
| Severe RVGE visits averted | 259 113 | 283 753 | 283 753 |
| Severe RVGE hospitalisations averted | 128 083 | 140 263 | 140 263 |
| Deaths averted | 19 752 | 21 631 | 21 631 |
| Healthcare cost averted (US$) | | | |
| Government perspective | 4 623 393 | 5 062 897 | 5 062 897 |
| Societal perspective | 7 045 141 | 7 714 859 | 7 714 859 |
| DALYs averted | 496 905 | 544 142 | 544 142 |
| Vaccine programme cost (US$) | 77 257 820 | 46 670 448 | 61 805 765 |
| Cost-effectiveness ratio (US$ per DALYs averted compared with no vaccine scenario) | | | |
| Government perspective | 146 | 76 | 104 |
| Societal perspective | 141 | 72 | 99 |

DALY, disability-adjusted life-year; RVGE, rotavirus gastroenteritis.

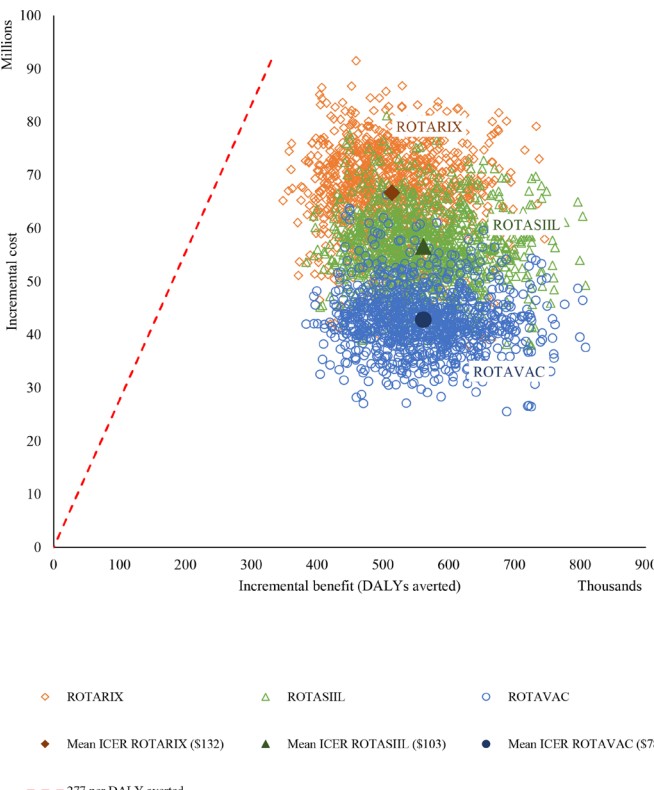

**Figure 1** Mean ICER and probabilistic cost-effectiveness results from the probabilistic uncertainty analysis, government perspective, over a 10-year period. ICER, incremental cost-effectiveness ratio.

Both ROTARIX and ROTASIIL may not be cost-effective given the availability of ROTAVAC; that is, if ROTARIX is compared directly to ROTAVAC, the cost per DALY averted would be US$638, and if ROTASIIL is compared directly to ROTAVAC, health benefits are the same, but vaccination programme costs are higher (ICER is undefined). However, there is considerable uncertainty around

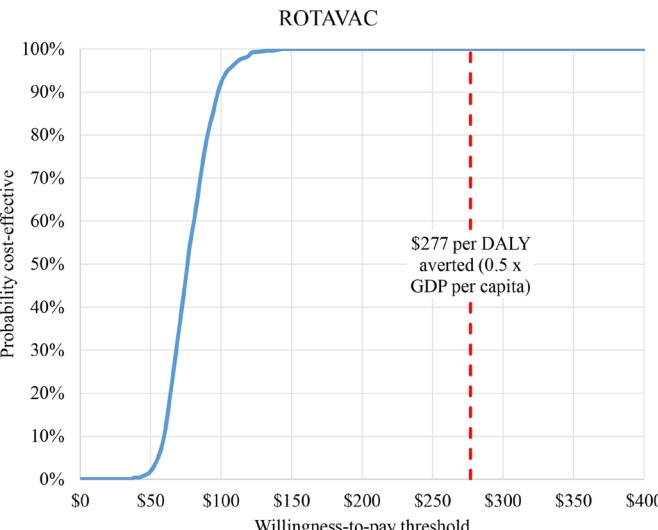

**Figure 2** Cost-effectiveness acceptability curve for the dominant option (ROTAVAC). DALY, disability-adjusted life-year; GDP, gross domestic product.

several of our inputs and the rank order would change if there were changes in important variables.

## Scenario analysis

The scenario analysis shows that all three rotavirus vaccines have cost-effectiveness ratios below the half GDP per capita threshold (online supplemental figure S1). The most favourable scenario is the scenario where we account for the level of co-financing currently supported by the country. When accounting for vaccine cofinancing instead of the Gavi price from the government perspective, ROTAVAC has a cost-effectiveness ratio of US$36, ROTASIIL US$59 and ROTARIX US$62. The least favourable scenario is the scenario where we account for a lower burden of disease. In this scenario, from the government perspective, the cost to avert a DALY is US$122 with ROTAVAC, US$165 with ROTASIIL and US$230 with ROTARIX.

## Probabilistic sensitivity analysis

Cost-effectiveness of rotavirus vaccination in Niger is further confirmed by the probabilistic uncertainty analysis whereby all 1000 runs returned cost-effectiveness ratio values below the assumed threshold. The same order of the vaccines' cost-effectiveness ratios remains, with ROTAVAC dominating other options (less costly and better health outcome) at a mean ICER of US$73 per DALY averted, ROTASIIL at US$103 and ROTARIX at US$132 (figure 1). The cost-effectiveness acceptability curve shows the high probablity for the dominant vaccine option to be cost-effective at thresholds lower than US$277. ROTAVAC has >99% chance of being cost-effective at a WTP threshold of US$150 (figure 2).

## DISCUSSION

Rotarix was officially included in the national immunisation programme in Niger in 2014, but no economic evaluation was conducted at the time nor widespread use. In 2014, only two vaccine presentations were available (ROTARIX and RotaTeq). At end of 2021, there are more than 10 rotavirus vaccine presentations that countries can choose from, making continued analysis essential to make informed choices.

This analysis confirms that rotavirus vaccine averts a substantial burden in Niger and remains cost-effective several years after introduction. All vaccines evaluated have ICERs below half of the GDP per capita WTP threshold, despite each product's differences in presentation and characteristics. Cost-effectiveness ratios for the vaccines are generally of similar scale, with ROTAVAC dominating other options. ROTAVAC's economic advantage is mainly driven by its lower volume resulting in lower immunisation delivery cost and lower price. This finding is consistent across all scenarios and probabilistic analyses and remains true even in the current context with Niger paying a share of the vaccine price through Gavi coprocurement. This study demonstrates that product

presentations and their associated costs should be evaluated closely.

There are limits to this analysis worth highlighting. The immunisation cost of delivery values used in the study are calculated based on the volume of each product. This advantages the product with the lowest cold chain footprint. Other factors should be considered when evaluating or revaluating product selection. In our example, policy-makers may wish to consider the feasibility of using a product with a smaller cold chain footprint that also must be stored in the negative cold chain at central level. ROTASIIL, though a higher volume product, could be, under certain circumstances, stored outside of the cold chain. If use of such a thermostable product was feasible country-wide in Niger without incurring additional costs, then ROTASIIL would potentially become the best choice from an economic perspective. Additional costs may be incurred from having to develop separate processes for an individual vaccine. Experiences to date with use of vaccines outside of the cold chain, through the Controlled Temperature Chain protocol, have led to positive outcomes in campaign settings. This approach is, however, not currently recommended for routine vaccines.[32]

The ROSE trial provided unique primary data on cost of delivery and households' cost of illness. Immunisation cost of delivery data are coming from a limited number of facilities, from one region only. While we address this limitation through sensitivity analysis, the immunisation delivery cost data would benefit from being based on a higher set of facilities and additional regions and districts. The same limitation applies to the cost of illness data used in the modelling. The small sample for hospitalised children was collected from a single facility within the ROSE trial.

Different impact values with two-dose versuss thee-dose vaccines are based on how the model calculates health outcomes based on efficacy values and waning of protection after the last dose of vaccine. The third dose of ROTAVAC and ROTASIIL is given 4 weeks later and thus translates in a small delay in the waning of protection compared with ROTARIX. This in turns results in a slightly higher number of averted cases, visits, hospitalisations and deaths. This has not been proven empirically, and studies that have assessed the impact of additional doses of vaccines have reported inconsistent results.[33] However, this is unlikely to impact our findings as, when accounting for similar impact among the three vaccines, the ROTARIX ICER remains inferior of that of ROTAVAC or ROTASIIL (US$133 per DALY averted from the government perspective). The limited data available on large-scale ROTAVAC and ROTASIIL effectiveness and safety data on these two vaccines, are two other factors that represent limits to our findings. A recent impact study in Palestine showed that benefits of rotavirus vaccination were sustained 2 years after a switch from ROTARIX to ROTAVAC.[34]

This evaluation does not account for indirect effects of vaccination. However, considering recent work by Park *et al*, it looks like indirect effects may not be a key driver in Niger and that our static model may provide reasonable results.[35]

Despite the involvement of national authorities and stakeholders, and use of locally collected data, there may be additional factors influencing decision-making that are not accounted for in the analysis.

The findings of this study may nevertheless contribute to informing decision-making around rotavirus vaccination policy in Niger, demonstrating that ROTAVAC is likely to be the most cost-effective option. Alternative products (ROTASIIL and ROTARIX) may also be considered by decision makers if they can be priced more competitively, or if their cold chain requirements could bring additional economic benefits. Regular re-evaluation is likely to be required as more evidence emerges on the relative costs and benefits of each product. Emerging data on the cost to switch vaccines should also be considered as a factor to account for when making decision about a change in product. Findings from few countries that have switched rotavirus vaccines in the past years are available and additional data are expected to become available with a number of countries planning to switch vaccines in 2022.[36 37]

**Contributors** FD (guarantor), KT, CP and SI designed the study. FD, KT, BA and OG collected and analysed the data. AC developed the study model. CP, AC, RFG and SI provided technical inputs to the study. FD ran the analysis and wrote the first draft of the manuscript. All authors reviewed and approved the final manuscript.

**Funding** This work was supported, in whole or in part, by the Bill & Melinda Gates Foundation (OPP1147721). Under the grant conditions of the Foundation, a Creative Commons Attribution 4.0 Generic License has already been assigned to the Author Accepted Manuscript version that might arise from this submission. ROSE trial funding: Funding was provided to Epicentre by Médecins Sans Frontières (MSF)-Operational Center Geneva (https://www.msf.org/) and the Kavli Foundation, Norway (https://www.kavlifoundation.org/). Serum Institute of India provided vaccine and placebo in kind.

**Competing interests** None declared.

**Patient and public involvement** Patients and/or the public were not involved in the design, or conduct, or reporting, or dissemination plans of this research.

**Patient consent for publication** Not applicable.

**Provenance and peer review** Not commissioned; externally peer reviewed.

**Data availability statement** All data relevant to the study are included in the article or uploaded as online supplemental information. The model used is available online at https://www.paho.org/en/provac-toolkit.

**ORCID iDs**
Frédéric Debellut http://orcid.org/0000-0002-3027-2838

Kevin Tang http://orcid.org/0000-0003-2580-3726

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
