## [Reviewer comments · BMJ Open]

ARTICLE DETAILS

TITLE (PROVISIONAL)	Impact and cost-effectiveness of rotavirus vaccination in Niger: a modeling study evaluating alternative rotavirus vaccines
AUTHORS	Debellut, Frédéric; Tang, Kevin; Clark, Andrew; Pecenka, Clint; Assao, Bachir; Guindo, Ousmane; Grais, RF; Isanaka, Sheila

VERSION 1 – REVIEW

REVIEWER	Burnett , Eleanor Centers for Disease Control and Prevention Prevention Research Centers
REVIEW RETURNED	20-Apr-2022

GENERAL COMMENTS	This is a well written and considered article on an important topic. Given the limited data to date for ROTAVAC and ROTASIIIL, Niger is a good choice for this evaluation. Given the number of figures and tables and the volume of data and scenarios you ran, the results section feels thin. Consider adding additional detailed results. Was there any attempt to calculate the costs of ROTASIIIL outside the cold chain? I didn't see this stated in the results. If not, this is a substantial portion of the discussion to not have any results. This is also the only aspect explicitly labeled as a limitation in the discussion. There are other limitations to the analysis, for example limited real world efficacy data for ROTAVAC and ROTASIIIL, lack of safety data specifically around intussusception for these 2 vaccines etc. Is there any information about the cost of Niger switching vaccines, given the Rotarix program is already established? This would be another point of consideration for decisionmakers. Move Gavi explanation from methods to intro. The paragraph beginning on line 292 seems out of place. Consider incorporating this point about the range of presentations elsewhere. Several of the figures and tables need additional information in their titles and labels. A few examples: Figure 2—what is the y-axis? Figure S1 needs a more detailed explanation/figure title. The difference between table 3 and table S4 is not clear (although it is stated in the results text). Additional information to distinguish the titles would help the reader.
---

REVIEWER	Malik, Akash United Nations Development Programme India
REVIEW RETURNED	10-Jun-2022

GENERAL COMMENTS	Comments for the paper titled "Evaluating the potential impact and cost-effectiveness of alternative rotavirus vaccines in Niger". Background – The background is very well described and establishes the need for such and assessment. It also describes the ROSE trial in detail from which this assessment has been branched out. Furthermore, the background also describes the three products being compared in the assessment which makes it easy for the reader to understand the context. Methodology – The methodology clearly highlights the models, variables and statistics that have been used for data analysis in this study. The tables clearly describe the variables that have been used for each of the 3 vaccine types. However, it would have been great if the authors could have further described in detail the cost of government perspective in societal perspective in terms of what factors were considered for each of this variable. Similarly for immunization delivery cost per dose ROSE costing study has been used for estimating the cost. It would be useful for the readers to have a brief understanding what program components were considered to estimate the delivery cost for example what percentage of the delivery cost attributed to cold chain, what percentage was attributed to actual vaccine delivery, similarly for communication, advocacy, training etc. This would help to the readers and the policymakers to have a clear understanding of what are the various programmatic components which are essential for introduction of rotavirus vaccine in a country. Results and Discussion – The results and discussion are in lines with the methodology. One observation is that the discussion only highlights the volume of a vaccine type as a limiting factor, however the fact that the study was based on a secondary data analysis without interaction with the health care workers, beneficiaries and policy makers has its own limitation and effect on data analysis and outcomes which need to be more clearly and confidently highlighted in relevant sections of the publication.
---

VERSION 1 – AUTHOR RESPONSE

Reviewer: 1

Comments to the Author:

This is a well written and considered article on an important topic. Given the limited data to date for ROTAVAC and ROTASIIL, Niger is a good choice for this evaluation.

Thank you for your assessment and review of our manuscript.

Given the number of figures and tables and the volume of data and scenarios you ran, the results section feels thin. Consider adding additional detailed results.

We added some results focusing on the annual cost of the vaccination program and the share of vaccines and supplies procurement:

“On average, undiscounted annual vaccination program costs in absence of Gavi support vary between US\$5.5 million with ROTAVAC and US\$8.9 million with ROTARIX, with the majority of the cost dedicated to vaccines and supplies procurement (55% for ROTAVAC, 60% with ROTARIX).” Was there any attempt to calculate the costs of ROTASIIL outside the cold chain? I didn’t see this stated in the results. If not, this is a substantial portion of the discussion to not have any results. We did not attempt to cost use of ROTASIIL outside of the cold chain because it is not currently used that way in Niger, and because it is unclear if the vaccine could be used outside of the cold chain in the entire country.

This is also the only aspect explicitly labeled as a limitation in the discussion. There are other limitations to the analysis, for example limited real world efficacy data for ROTAVAC and ROTASIIL, lack of safety data specifically around intussusception for these 2 vaccines etc.

We added these suggestions as limitations to the study:

“The limited data available on large-scale ROTAVAC and ROTASIIL effectiveness and safety data on these two vaccines, are two other factors that represent limits to our findings. A recent impact study in Palestine showed that benefits of rotavirus vaccination were sustained two years after a switch from ROTARIX to ROTAVAC.”ⁱ

ⁱ *Palestinian Ministry of Health, Rostropovich Vishnevskaya Foundation, PATH. The health benefits of rotavirus immunization for children in Palestine Results of a vaccine impact analysis. Brief published Sept. 2021. Available from*

https://path.azureedge.net/media/documents/PATH_Palestine_Rotavirus_Epidemiology_ENGLISH_0d0BYfV.pdf. Accessed on 23 June 2022.

We also highlight additional limitations to the study regarding the use of locally collected data with rather small sample from a single region and the use of a static model not allowing to account for potential indirect effects.

Is there any information about the cost of Niger switching vaccines, given the Rotarix program is already established? This would be another point of consideration for decisionmakers.

There is no information on potential cost to switch vaccines in Niger, however, some data was generated from other countries that recently switched rotavirus vaccines and considering the number of countries that are scheduled to switch in 2022, more information should be available in the future. We added this as a consideration to policy makers at the end of the discussion section and provide references to available data:

“Emerging data on the cost to switch vaccines should also be considered as a factor to account for when making decision about a change in product. Findings from few countries that have switched rotavirus vaccines in the past years are available and additional data is expected to become available with a number of countries planning to switch vaccines in 2022.”^{i, ii}

ⁱ *Debellut F, Jaber S, Bouzya Y, Sabbah J, Barham M, Abu-Awwad F, et al. Introduction of rotavirus vaccination in Palestine: An evaluation of the costs, impact, and cost-effectiveness of ROTARIX and ROTAVAC. PLoS One 2020;15:e0228506. <https://doi.org/10.1371/JOURNAL.PONE.0228506>.*

ⁱⁱ *PATH, The switch from ROTARIX to ROTAVAC in Ghana: Answers to four key questions. Available from <https://www.path.org/resources/switch-rotarix-rotavac-ghana-answers-four-key-questions/>. Accessed on 23 June 2022.*

Move Gavi explanation from methods to intro.

We moved the sentence “Gavi is an international organization aiming to increase equitable and sustainable access to vaccines for low- and middle-income countries. As a result, countries eligible for Gavi support can access negotiated prices from manufacturers through procurement with UNICEF” from the methods to the introduction section.

The paragraph beginning on line 292 seems out of place. Consider incorporating this point about the range of presentations elsewhere.

We moved the following paragraph to the beginning of the discussion section as this represents a good rationale for the study.

“Rotarix was officially included in the national immunization program in Niger in 2014, but no economic evaluation was conducted at the time nor widespread use. In 2014 only two vaccine presentations were available (ROTARIX and RotaTeq). At end of 2021, there are more than 10 rotavirus vaccine presentations that countries can choose from, making continued analysis essential to make informed choices.”

Several of the figures and tables need additional information in their titles and labels. A few examples: Figure 2—what is the y-axis?

We added the label for Figure 2's Y axis: “Probability cost-effective”

Figure S1 needs a more detailed explanation/figure title.

We revised the title of the figure to: “Figure S 1: Scenario analysis results showing incremental cost-effectiveness ratio (US\$ per DALY averted) from the government and societal perspectives of ROTARIX, ROTAVAC, and ROTASIIL compared to no vaccination.”

The difference between table 3 and table S4 is not clear (although it is stated in the results text).

Additional information to distinguish the titles would help the reader.

We revised the title of table S4 to: “Table S4: Health and economic benefits of vaccine (2021-2030, discounted), assuming 2-dose ROTARIX had similar impact to the 3-dose ROTAVAC or ROTASIIL.”

Reviewer: 2

Comments to the Author:

The paper submitted to BMJ for review aims to determine the cost-effectiveness of the introduction of Rotavirus vaccine products available through GAVI i.e. ROTACIIL, ROTAVAC and ROTARIX. All the three products have been introduced in different GAVI-funded countries and have shown an impact in terms of reduction of burden of Rotavirus diarrheal disease. Through this paper the policy makers in Niger as well as other similar countries will be able to take an informed decision for which product will be most effective in terms of cost and DALY for their respective countries. I would like to congratulate the authors on the excellent presentation, simple explanation and a very informative discussion.

Many thanks for your review of our manuscript and note of appreciation.

Comments for the paper titled “Evaluating the potential impact and cost-effectiveness of alternative rotavirus vaccines in Niger”.

Background – The background is very well described and establishes the need for such an assessment. It also describes the ROSE trial in detail from which this assessment has been branched out. Furthermore, the background also describes the three products being compared in the assessment which makes it easy for the reader to understand the context.

Methodology – The methodology clearly highlights the models, variables and statistics that have been used for data analysis in this study. The tables clearly describe the variables that have been used for each of the 3 vaccine types.

Many thanks for your review of our manuscript and note of appreciation.

However, it would have been great if the authors could have further described in detail the cost of government perspective in societal perspective in terms of what factors were considered for each of these variables.

We added definitions of both perspectives at the beginning of the methods section:

“The government perspective includes healthcare costs supported by the government only, while the societal perspective also accounts for healthcare costs supported by households.”

This short description comes in addition to the details provided further below in the methods:

“For the government perspective, we used estimates of direct medical costs of diarrhea as published by Baral *et al.* For the societal perspective, direct medical costs were combined with household costs

collected from a sub-group of caregivers presenting to ROSE trial health facilities with a child sick with RVGE. Household costs included out-of-pocket expenditures to seek care in terms of medicines, transportation, and food as well as indirect costs from time spent seeking care and caring for sick children. Caregiver time was valued using the annual GDP per capita divided by 365 days and multiplied by the number of days spent seeking care or caring for a child in hospital.”

Similarly for immunization delivery cost per dose ROSE costing study has been used for estimating the cost. It would be useful for the readers to have a brief understanding what program components were considered to estimate the delivery cost for example what percentage of the delivery cost attributed to cold chain, what percentage was attributed to actual vaccine delivery, similarly for communication, advocacy, training etc. This would help to the readers and the policymakers to have a clear understanding of what are the various programmatic components which are essential for introduction of rotavirus vaccine in a country.

We provide the details of immunization delivery cost for each vaccine and cost categories in supplementary Table S1. Because the proportion of each cost category varies depending on the vaccine of interest, we elected not to provide such detail in the paper but rather provide the numbers in supplement. However, in response to the reviewer’s comment, we have added the following sentence to provide some more information to readers:

“The main driver of cost was service delivery at health facility level, which represented between 30% and 60% of the total immunization delivery cost, depending on which vaccine was considered.”

Results and Discussion – The results and discussion are in lines with the methodology. One observation is that the discussion only highlights the volume of a vaccine type as a limiting factor, however the fact that the study was based on a secondary data analysis without interaction with the health care workers, beneficiaries and policy makers has its own limitation and effect on data analysis and outcomes which need to be more clearly and confidently highlighted in relevant sections of the publication.

Local and national authorities were engaged and informed of all aspects of the parent trial, including the costing data collection as well as at the origin of the trial itself. Local authorities were further involved in the dissemination of any pertinent findings of this analysis. Further to this, beneficiaries were consulted via a community advisory board. We recognize that even with this, the model itself does not include qualitative information or other variables which may weigh in on decision-making. This is now noted in the discussion:

“Despite the involvement of national authorities and stakeholders, and use of locally collected data, there may be additional factors influencing decision-making that are not accounted for in the analysis”.

VERSION 2 – REVIEW

REVIEWER	Malik, Akash United Nations Development Programme India
REVIEW RETURNED	04-Jul-2022
GENERAL COMMENTS	Thanks for sharing the supplemental data files and addressing the inputs at relevant places in the manuscript. Congratulations again for an excellent and informative policy level research.